# Avoiding Discrimination through Causal Reasoning

**Niki Kilbertus**[†‡]
nkilbertus@tue.mpg.de

**Mateo Rojas-Carulla**[†‡]
mrojas@tue.mpg.de

**Giambattista Parascandolo**[†§]
gparascandolo@tue.mpg.de

**Moritz Hardt**[*]
hardt@berkeley.edu

**Dominik Janzing**[†]
janzing@tue.mpg.de

**Bernhard Schölkopf**[†]
bs@tue.mpg.de

[†]Max Planck Institute for Intelligent Systems
[‡]University of Cambridge
[§]Max Planck ETH Center for Learning Systems
[*]University of California, Berkeley

## Abstract

Recent work on fairness in machine learning has focused on various statistical discrimination criteria and how they trade off. Most of these criteria are observational: They depend only on the joint distribution of predictor, protected attribute, features, and outcome. While convenient to work with, observational criteria have severe inherent limitations that prevent them from resolving matters of fairness conclusively.

Going beyond observational criteria, we frame the problem of discrimination based on protected attributes in the language of causal reasoning. This viewpoint shifts attention from "What is the right fairness criterion?" to "What do we want to assume about our model of the causal data generating process?" Through the lens of causality, we make several contributions. First, we crisply articulate why and when observational criteria fail, thus formalizing what was before a matter of opinion. Second, our approach exposes previously ignored subtleties and why they are fundamental to the problem. Finally, we put forward natural causal non-discrimination criteria and develop algorithms that satisfy them.

## 1   Introduction

As machine learning progresses rapidly, its societal impact has come under scrutiny. An important concern is potential discrimination based on protected attributes such as gender, race, or religion. Since learned predictors and risk scores increasingly support or even replace human judgment, there is an opportunity to formalize what harmful discrimination means and to design algorithms that avoid it. However, researchers have found it difficult to agree on a single measure of discrimination. As of now, there are several competing approaches, representing different opinions and striking different trade-offs. Most of the proposed fairness criteria are observational: They depend only on the joint distribution of predictor $R$, protected attribute $A$, features $X$, and outcome $Y$. For example, the natural requirement that $R$ and $A$ must be statistically independent is referred to as *demographic parity*. Some approaches transform the features $X$ to obfuscate the information they contain about $A$ [1]. The recently proposed *equalized odds* constraint [2] demands that the predictor $R$ and the attribute $A$ be independent conditional on the actual outcome $Y$. All three are examples of observational approaches.

A growing line of work points at the insufficiency of existing definitions. Hardt, Price and Srebro [2] construct two scenarios with *intuitively* different social interpretations that admit identical joint dis-

tributions over $(R, A, Y, X)$. Thus, no observational criterion can distinguish them. While there are non-observational criteria, notably the early work on individual fairness [3], these have not yet gained traction. So, it might appear that the community has reached an impasse.

## 1.1 Our contributions

We assay the problem of discrimination in machine learning in the language of causal reasoning. This viewpoint supports several contributions:

- Revisiting the two scenarios proposed in [2], we articulate a natural causal criterion that formally distinguishes them. In particular, we show that observational criteria are unable to determine if a protected attribute has *direct causal influence* on the predictor that is not mitigated by *resolving* variables.

- We point out subtleties in fair decision making that arise naturally from a causal perspective, but have gone widely overlooked in the past. Specifically, we formally argue for the need to distinguish between the underlying concept behind a protected attribute, such as race or gender, and its *proxies* available to the algorithm, such as visual features or name.

- We introduce and discuss two natural causal criteria centered around the notion of *interventions* (relative to a causal graph) to formally describe specific forms of discrimination.

- Finally, we initiate the study of algorithms that avoid these forms of discrimination. Under certain linearity assumptions about the underlying causal model generating the data, an algorithm to remove a specific kind of discrimination leads to a simple and natural heuristic.

At a higher level, our work proposes a shift from trying to find a single statistical fairness criterion to arguing about properties of the data and which assumptions about the generating process are justified. Causality provides a flexible framework for organizing such assumptions.

## 1.2 Related work

Demographic parity and its variants have been discussed in numerous papers, e.g., [1, 4–6]. While demographic parity is easy to work with, the authors of [3] already highlighted its insufficiency as a fairness constraint. In an attempt to remedy the shortcomings of demographic parity [2] proposed two notions, *equal opportunity* and *equal odds*, that were also considered in [7]. A review of various fairness criteria can be found in [8], where they are discussed in the context of criminal justice. In [9, 10] it has been shown that imperfect predictors cannot simultaneously satisfy equal odds and *calibration* unless the groups have identical base rates, i.e. rates of positive outcomes.

A starting point for our investigation is the unidentifiability result of [2]. It shows that observational criteria are too weak to distinguish two intuitively very different scenarios. However, the work does not provide a formal mechanism to articulate why and how these scenarios should be considered different. Inspired by Pearl's causal interpretation of Simpson's paradox [11, Section 6], we propose causality as a way of coping with this unidentifiability result.

An interesting non-observational fairness definition is the notion of *individual fairness* [3] that assumes the existence of a similarity measure on individuals, and requires that any two similar individuals should receive a similar distribution over outcomes. More recent work lends additional support to such a definition [12]. From the perspective of causality, the idea of a similarity measure is akin to the method of *matching* in counterfactual reasoning [13, 14]. That is, evaluating approximate counterfactuals by comparing individuals with similar values of covariates excluding the protected attribute.

Recently, [15] put forward one possible causal definition, namely the notion of *counterfactual fairness*. It requires modeling counterfactuals on a per individual level, which is a delicate task. Even determining the effect of *race* at the group level is difficult; see the discussion in [16]. The goal of our paper is to assay a more general causal framework for reasoning about discrimination in machine learning without committing to a single fairness criterion, and without committing to evaluating individual causal effects. In particular, we draw an explicit distinction between the protected attribute (for which interventions are often impossible in practice) and its proxies (which sometimes can be intervened upon).

Moreover, causality has already been employed for the discovery of discrimination in existing data sets by [14, 17]. Causal graphical conditions to identify *meaningful partitions* have been proposed for the discovery and prevention of certain types of discrimination by preprocessing the data [18]. These conditions rely on the evaluation of *path specific effects*, which can be traced back all the way to [11, Section 4.5.3]. The authors of [19] recently picked up this notion and generalized Pearl's approach by a constraint based prevention of discriminatory path specific effects arising from counterfactual reasoning. Our research was done independently of these works.

## 1.3 Causal graphs and notation

Causal graphs are a convenient way of organizing assumptions about the data generating process. We will generally consider causal graphs involving a protected attribute $A$, a set of proxy variables $P$, features $X$, a predictor $R$ and sometimes an observed outcome $Y$. For background on causal graphs see [11]. In the present paper a *causal graph* is a directed, acyclic graph whose nodes represent random variables. A *directed path* is a sequence of distinct nodes $V_1, \ldots, V_k$, for $k \geq 2$, such that $V_i \to V_{i+1}$ for all $i \in \{1, \ldots, k-1\}$. We say a directed path is *blocked by a set of nodes $Z$*, where $V_1, V_k \notin Z$, if $V_i \in Z$ for some $i \in \{2, \ldots, k-1\}$.[1]

A *structural equation model* is a set of equations $V_i = f_i(pa(V_i), N_i)$, for $i \in \{1, \ldots, n\}$, where $pa(V_i)$ are the parents of $V_i$, i.e. its *direct causes*, and the $N_i$ are independent noise variables. We interpret these equations as assignments. Because we assume acyclicity, starting from the roots of the graph, we can recursively compute the other variables, given the noise variables. This leads us to view the structural equation model and its corresponding graph as a *data generating model*. The predictor $R$ maps inputs, e.g., the features $X$, to a predicted output. Hence we model it as a childless node, whose parents are its input variables. Finally, note that given the noise variables, a structural equation model entails a unique joint distribution; however, the same joint distribution can usually be entailed by multiple structural equation models corresponding to distinct causal structures.

# 2 Unresolved discrimination and limitations of observational criteria

To bear out the limitations of observational criteria, we turn to Pearl's commentary on claimed gender discrimination in Berkeley college admissions [11, Section 4.5.3]. Bickel [20] had shown earlier that a lower college-wide admission rate for women than for men was explained by the fact that women applied in more competitive departments. When adjusted for department choice, women experienced a slightly higher acceptance rate compared with men. From the causal point of view, what matters is the *direct effect* of the protected attribute (here, gender $A$) on the decision (here, college admission $R$) that cannot be ascribed to a *resolving variable* such as department choice $X$, see Figure 1. We shall use the term *resolving variable* for any variable in the causal graph that is influenced by $A$ in a manner that we accept as non-discriminatory. With this convention, the criterion can be stated as follows.

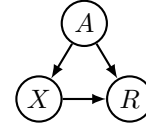

Figure 1: The admission decision $R$ does not only directly depend on gender $A$, but also on department choice $X$, which in turn is also affected by gender $A$.

**Definition 1** (Unresolved discrimination). A variable $V$ in a causal graph exhibits *unresolved discrimination* if there exists a directed path from $A$ to $V$ that is not blocked by a resolving variable and $V$ itself is non-resolving.

Pearl's commentary is consistent with what we call the *skeptic viewpoint*. All paths from the protected attribute $A$ to $R$ are problematic, unless they are justified by a resolving variable. The presence of unresolved discrimination in the predictor $R$ is worrisome and demands further scrutiny. In practice, $R$ is not a priori part of a given graph. Instead it is our objective to construct it as a function of the features $X$, some of which might be resolving. Hence we should first look for unresolved discrimination in the features. A canonical way to avoid unresolved discrimination in $R$ is to only input the set of features that do not exhibit unresolved discrimination. However, the remaining

features might be affected by non-resolving *and* resolving variables. In Section 4 we investigate whether one can exclusively remove unresolved discrimination from such features. A related notion of "explanatory features" in a non-causal setting was introduced in [21].

The definition of unresolved discrimination in a predictor has some interesting special cases worth highlighting. If we take the set of resolving variables to be empty, we intuitively get a causal analog of demographic parity. No directed paths from $A$ to $R$ are allowed, but $A$ and $R$ can still be statistically dependent. Similarly, if we choose the set of resolving variables to be the singleton set $\{Y\}$ containing the true outcome, we obtain a causal analog of equalized odds where strict independence is not necessary. The causal intuition implied by "the protected attribute should not affect the prediction", and "the protected attribute can only affect the prediction when the information comes through the true label", is neglected by (conditional) statistical independences $A \perp\!\!\!\perp R$, and $A \perp\!\!\!\perp R \,|\, Y$, but well captured by only considering dependences mitigated along directed causal paths.

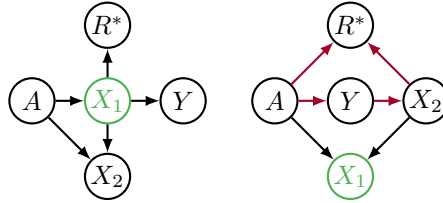

Figure 2: Two graphs that may generate the same joint distribution for the Bayes optimal unconstrained predictor $R^*$. If $X_1$ is a resolving variable, $R^*$ exhibits unresolved discrimination in the right graph (along the red paths), but not in the left one.

We will next show that observational criteria are fundamentally unable to determine whether a predictor exhibits unresolved discrimination or not. This is true even if the predictor is *Bayes optimal*. In passing, we also note that fairness criteria such as equalized odds may or may not exhibit unresolved discrimination, but this is again something an observational criterion cannot determine.

**Theorem 1.** *Given a joint distribution over the protected attribute $A$, the true label $Y$, and some features $X_1, \ldots, X_n$, in which we have already specified the resolving variables, no observational criterion can generally determine whether the Bayes optimal unconstrained predictor or the Bayes optimal equal odds predictor exhibit unresolved discrimination.*

All proofs for the statements in this paper are in the supplementary material.

The two graphs in Figure 2 are taken from [2], which we here reinterpret in the causal context to prove Theorem 1. We point out that there is an established set of conditions under which unresolved discrimination can, in fact, be determined from observational data. Note that the two graphs are not Markov equivalent. Therefore, to obtain the same joint distribution we must violate a condition called *faithfulness*.[2] We later argue that violation of faithfulness is by no means pathological, but emerges naturally when designing predictors. In any case, interpreting conditional dependences can be difficult in practice [22].

## 3 Proxy discrimination and interventions

We now turn to an important aspect of our framework. Determining causal effects in general requires modeling interventions. Interventions on deeply rooted individual properties such as *gender* or *race* are notoriously difficult to conceptualize—especially at an individual level, and impossible to perform in a randomized trial. VanderWeele et al. [16] discuss the problem comprehensively in an epidemiological setting. From a machine learning perspective, it thus makes sense to separate the protected attribute $A$ from its potential *proxies*, such as name, visual features, languages spoken at home, etc. Intervention based on proxy variables poses a more manageable problem. By deciding on a suitable proxy we can find an adequate mounting point for determining and removing its influence on the prediction. Moreover, in practice we are often limited to imperfect measurements of $A$ in any case, making the distinction between root concept and proxy prudent.

As was the case with resolving variables, a *proxy* is a priori nothing more than a descendant of $A$ in the causal graph that we choose to label as a proxy. Nevertheless in reality we envision the proxy

to be a clearly defined observable quantity that is significantly correlated with $A$, yet in our view should not affect the prediction.

**Definition 2** (Potential proxy discrimination). A variable $V$ in a causal graph exhibits *potential proxy discrimination*, if there exists a directed path from $A$ to $V$ that is blocked by a proxy variable and $V$ itself is not a proxy.

Potential proxy discrimination articulates a causal criterion that is in a sense dual to unresolved discrimination. From the *benevolent viewpoint*, we *allow* any path from $A$ to $R$ unless it passes through a proxy variable, which we consider worrisome. This viewpoint acknowledges the fact that the influence of $A$ on the graph may be complex and it can be too restraining to rule out all but a few designated features. In practice, as with unresolved discrimination, we can naively build an unconstrained predictor based only on those features that do not exhibit potential proxy discrimination. Then we must not provide $P$ as input to $R$; unawareness, i.e. excluding $P$ from the inputs of $R$, suffices. However, by granting $R$ access to $P$, we can carefully tune the function $R(P, X)$ to cancel the implicit influence of $P$ on features $X$ that exhibit potential proxy discrimination by the explicit dependence on $P$. Due to this possible cancellation of paths, we called the path based criterion *potential* proxy discrimination. When building predictors that exhibit no *overall proxy discrimination*, we precisely aim for such a cancellation.

Fortunately, this idea can be conveniently expressed by an *intervention* on $P$, which is denoted by $do(P = p)$ [11]. Visually, intervening on $P$ amounts to removing all incoming arrows of $P$ in the graph; algebraically, it consists of replacing the structural equation of $P$ by $P = p$, i.e. we put point mass on the value $p$.

**Definition 3** (Proxy discrimination). A predictor $R$ exhibits no *proxy discrimination* based on a proxy $P$ if for all $p, p'$

$$\mathbb{P}(R \,|\, do(P = p)) = \mathbb{P}(R \,|\, do(P = p')) \,. \tag{1}$$

The interventional characterization of proxy discrimination leads to a simple procedure to remove it in causal graphs that we will turn to in the next section. It also leads to several natural variants of the definition that we discuss in Section 4.3. We remark that Equation (1) is an equality of probabilities in the "do-calculus" that cannot in general be inferred by an observational method, because it depends on an underlying causal graph, see the discussion in [11]. However, in some cases, we do not need to resort to interventions to avoid proxy discrimination.

**Proposition 1.** *If there is no directed path from a proxy to a feature, unawareness avoids proxy discrimination.*

## 4   Procedures for avoiding discrimination

Having motivated the two types of discrimination that we distinguish, we now turn to building predictors that avoid them in a given causal model. First, we remark that a more comprehensive treatment requires individual judgement of not only variables, but the legitimacy of every existing path that ends in $R$, i.e. evaluation of *path-specific effects* [18, 19], which is tedious in practice. The natural concept of proxies and resolving variables covers most relevant scenarios and allows for natural removal procedures.

### 4.1   Avoiding proxy discrimination

While presenting the general procedure, we illustrate each step in the example shown in Figure 3. A protected attribute $A$ affects a proxy $P$ as well as a feature $X$. Both $P$ and $X$ have additional unobserved causes $N_P$ and $N_X$, where $N_P, N_X, A$ are pairwise independent. Finally, the proxy also has an effect on the features $X$ and the predictor $R$ is a function of $P$ and $X$. Given labeled training data, our task is to find a good predictor that exhibits no proxy discrimination within a hypothesis class of functions $R_\theta(P, X)$ parameterized by a real valued vector $\theta$.

We now work out a formal procedure to solve this task under specific assumptions and simultaneously illustrate it in a fully linear example, i.e. the structural equations are given by

$$P = \alpha_P A + N_P, \qquad X = \alpha_X A + \beta P + N_X, \qquad R_\theta = \lambda_P P + \lambda_X X \,.$$

Note that we choose linear functions parameterized by $\theta = (\lambda_P, \lambda_X)$ as the hypothesis class for $R_\theta(P, X)$.

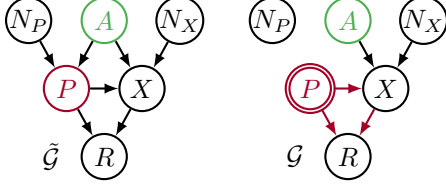
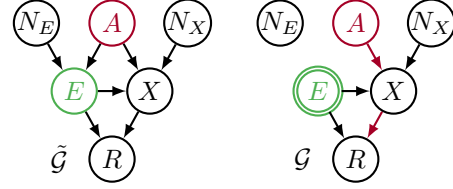

Figure 3: A template graph $\tilde{\mathcal{G}}$ for proxy discrimination (left) with its intervened version $\mathcal{G}$ (right). While from the benevolent viewpoint we do not generically prohibit any influence from $A$ on $R$, we want to guarantee that the proxy $P$ has no overall influence on the prediction, by adjusting $P \to R$ to cancel the influence along $P \to X \to R$ in the intervened graph.

Figure 4: A template graph $\tilde{\mathcal{G}}$ for unresolved discrimination (left) with its intervened version $\mathcal{G}$ (right). While from the skeptical viewpoint we generically do not want $A$ to influence $R$, we first intervene on $E$ interrupting all paths through $E$ and only cancel the remaining influence on $A$ to $R$.

We will refer to the *terminal ancestors of a node $V$ in a causal graph $\mathcal{D}$*, denoted by $ta^{\mathcal{D}}(V)$, which are those ancestors of $V$ that are also root nodes of $\mathcal{D}$. Moreover, in the procedure we clarify the notion of *expressibility*, which is an assumption about the relation of the given structural equations and the hypothesis class we choose for $R_\theta$.

**Proposition 2.** *If there is a choice of parameters $\theta_0$ such that $R_{\theta_0}(P, X)$ is constant with respect to its first argument and the structural equations are* expressible*, the following procedure returns a predictor from the given hypothesis class that exhibits no proxy discrimination and is non-trivial in the sense that it can make use of features that exhibit potential proxy discrimination.*

1. Intervene on $P$ by removing all incoming arrows and replacing the structural equation for $P$ by $P = p$. For the example in Figure 3,

$$P = p, \qquad X = \alpha_X A + \beta P + N_X, \qquad R_\theta = \lambda_P P + \lambda_X X. \tag{2}$$

2. Iteratively substitute variables in the equation for $R_\theta$ from their structural equations until only root nodes of the intervened graph are left, i.e. write $R_\theta(P, X)$ as $R_\theta(P, g(ta^{\mathcal{G}}(X)))$ for some function $g$. In the example, $ta(X) = \{A, P, N_X\}$ and

$$R_\theta = (\lambda_P + \lambda_X \beta)p + \lambda_X(\alpha_X A + N_X). \tag{3}$$

3. We now require the distribution of $R_\theta$ in (3) to be independent of $p$, i.e. for all $p, p'$

$$\mathbb{P}((\lambda_P + \lambda_X \beta)p + \lambda_X(\alpha_X A + N_X)) = \mathbb{P}((\lambda_P + \lambda_X \beta)p' + \lambda_X(\alpha_X A + N_X)). \tag{4}$$

We seek to write the predictor as a function of $P$ and all the other roots of $\mathcal{G}$ separately. If our hypothesis class is such that there exists $\tilde{\theta}$ such that $R_\theta(P, g(ta(X))) = R_{\tilde{\theta}}(P, \tilde{g}(ta(X) \setminus \{P\}))$, we call the structural equation model and hypothesis class specified in (2) *expressible*. In our example, this is possible with $\tilde{\theta} = (\lambda_P + \lambda_X \beta, \lambda_X)$ and $\tilde{g} = \alpha_X A + N_X$. Equation (4) then yields the *non-discrimination constraint* $\tilde{\theta} = \theta_0$. Here, a possible $\theta_0$ is $\theta_0 = (0, \lambda_X)$, which simply yields $\lambda_P = -\lambda_X \beta$.

4. Given labeled training data, we can optimize the predictor $R_\theta$ within the hypothesis class as given in (2), subject to the non-discrimination constraint. In the example

$$R_\theta = -\lambda_X \beta P + \lambda_X X = \lambda_X(X - \beta P),$$

with the free parameter $\lambda_X \in \mathbb{R}$.

In general, the non-discrimination constraint (4) is by construction just $\mathbb{P}(R \,|\, do(P = p)) = \mathbb{P}(R \,|\, do(P = p'))$, coinciding with Definition 3. Thus Proposition 2 holds by construction of the procedure. The choice of $\theta_0$ strongly influences the non-discrimination constraint. However, as the example shows, it allows $R_\theta$ to exploit features that exhibit potential proxy discrimination.

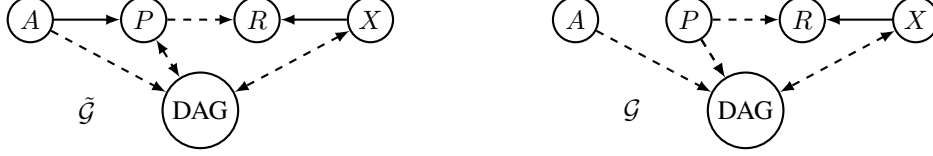

Figure 5: *Left:* A generic graph $\tilde{\mathcal{G}}$ to describe proxy discrimination. *Right:* The graph corresponding to an intervention on $P$. The circle labeled "DAG" represents any sub-DAG of $\tilde{\mathcal{G}}$ and $\mathcal{G}$ containing an arbitrary number of variables that is compatible with the shown arrows. Dashed arrows can, but do not have to be present in a given scenario.

## 4.2 Avoiding unresolved discrimination

We proceed analogously to the previous subsection using the example graph in Figure 4. Instead of the proxy, we consider a resolving variable $E$. The causal dependences are equivalent to the ones in Figure 3 and we again assume linear structural equations

$$E = \alpha_E A + N_E, \qquad X = \alpha_X A + \beta E + N_X, \qquad R_\theta = \lambda_E E + \lambda_X X .$$

Let us now try to adjust the previous procedure to the context of avoiding unresolved discrimination.

1. Intervene on $E$ by fixing it to a random variable $\eta$ with $\mathbb{P}(\eta) = \mathbb{P}(E)$, the marginal distribution of $E$ in $\tilde{\mathcal{G}}$, see Figure 4. In the example we find

$$E = \eta, \qquad X = \alpha_X A + \beta E + N_X, \qquad R_\theta = \lambda_E E + \lambda_X X . \qquad (5)$$

2. By iterative substitution write $R_\theta(E, X)$ as $R_\theta(E, g(ta^{\mathcal{G}}(X)))$ for some function $g$, i.e. in the example

$$R_\theta = (\lambda_E + \lambda_X \beta)\eta + \lambda_X \alpha_X A + \lambda_X N_X . \qquad (6)$$

3. We now demand the distribution of $R_\theta$ in (6) be invariant under interventions on $A$, which coincides with conditioning on $A$ whenever $A$ is a root of $\tilde{\mathcal{G}}$. Hence, in the example, for all $a, a'$

$$\mathbb{P}((\lambda_E + \lambda_X \beta)\eta + \lambda_X \alpha_X a + \lambda_X N_X)) = \mathbb{P}((\lambda_E + \lambda_X \beta)\eta + \lambda_X \alpha_X a' + \lambda_X N_X)) . \qquad (7)$$

Here, the subtle asymmetry between proxy discrimination and unresolved discrimination becomes apparent. Because $R_\theta$ is not explicitly a function of $A$, we cannot cancel implicit influences of $A$ through $X$. There might still be a $\theta_0$ such that $R_{\theta_0}$ indeed fulfils (7), but there is no principled way for us to construct it. In the example, (7) suggests the obvious *non-discrimination constraint* $\lambda_X = 0$. We can then proceed as before and, given labeled training data, optimize $R_\theta = \lambda_E E$ by varying $\lambda_E$. However, by setting $\lambda_X = 0$, we also cancel the path $A \to E \to X \to R$, even though it is blocked by a resolving variable. In general, if $R_\theta$ does not have access to $A$, we can not adjust for unresolved discrimination without also removing resolved influences from $A$ on $R_\theta$.

If, however, $R_\theta$ is a function of $A$, i.e. we add the term $\lambda_A A$ to $R_\theta$ in (5), the non-discrimination constraint is $\lambda_A = -\lambda_X \alpha_X$ and we can proceed analogously to the procedure for proxies.

## 4.3 Relating proxy discriminations to other notions of fairness

Motivated by the algorithm to avoid proxy discrimination, we discuss some natural variants of the notion in this section that connect our interventional approach to individual fairness and other proposed criteria. We consider a generic graph structure as shown on the left in Figure 5. The proxy $P$ and the features $X$ could be multidimensional. The empty circle in the middle represents any number of variables forming a DAG that respects the drawn arrows. Figure 3 is an example thereof. All dashed arrows are optional depending on the specifics of the situation.

**Definition 4.** A predictor $R$ exhibits no *individual proxy discrimination*, if for all $x$ and all $p, p'$

$$\mathbb{P}(R \,|\, do(P = p), X = x) = \mathbb{P}(R \,|\, do(P = p'), X = x) .$$

A predictor $R$ exhibits no *proxy discrimination in expectation*, if for all $p, p'$

$$\mathbb{E}[R \,|\, do(P = p)] = \mathbb{E}[R \,|\, do(P = p')] .$$

Individual proxy discrimination aims at comparing examples with the same features $X$, for different values of $P$. Note that this can be individuals with different values for the unobserved non-feature variables. A true individual-level comparison of the form "What would have happened to me, if I had always belonged to another group" is captured by *counterfactuals* and discussed in [15, 19].

For an analysis of proxy discrimination, we need the structural equations for $P, X, R$ in Figure 5

$$P = \hat{f}_P(pa(P)),$$
$$X = \hat{f}_X(pa(X)) = f_X(P, ta^{\mathcal{G}}(X) \setminus \{P\}),$$
$$R = \hat{f}_R(P, X) = f_R(P, ta^{\mathcal{G}}(R) \setminus \{P\}).$$

For convenience, we will use the notation $ta_P^{\mathcal{G}}(X) := ta^{\mathcal{G}}(X) \setminus \{P\}$. We can find $f_X, f_R$ from $\hat{f}_X, \hat{f}_R$ by first rewriting the functions in terms of root nodes of the *intervened graph*, shown on the right side of Figure 5, and then assigning the *overall* dependence on $P$ to the first argument.

We now compare proxy discrimination to other existing notions.

**Theorem 2.** *Let the influence of $P$ on $X$ be additive and linear, i.e.*

$$X = f_X(P, ta_P^{\mathcal{G}}(X)) = g_X(ta_P^{\mathcal{G}}(X)) + \mu_X P$$

*for some function $g_X$ and $\mu_X \in \mathbb{R}$. Then any predictor of the form*

$$R = r(X - \mathbb{E}[X \mid do(P)])$$

*for some function $r$ exhibits no proxy discrimination.*

Note that in general $\mathbb{E}[X \mid do(P)] \neq \mathbb{E}[X \mid P]$. Since in practice we only have observational data from $\tilde{\mathcal{G}}$, one cannot simply build a predictor based on the "regressed out features" $\tilde{X} := X - \mathbb{E}[X \mid P]$ to avoid proxy discrimination. In the scenario of Figure 3, the direct effect of $P$ on $X$ along the arrow $P \to X$ in the left graph cannot be estimated by $\mathbb{E}[X \mid P]$, because of the common confounder $A$. The desired interventional expectation $\mathbb{E}[X \mid do(P)]$ coincides with $\mathbb{E}[X \mid P]$ only if one of the arrows $A \to P$ or $A \to X$ is not present. Estimating direct causal effects is a hard problem, well studied by the causality community and often involves instrumental variables [23].

This cautions against the natural idea of using $\tilde{X}$ as a "fair representation" of $X$, as it implicitly neglects that we often want to remove the effect of proxies and not the protected attribute. Nevertheless, the notion agrees with our interventional proxy discrimination in some cases.

**Corollary 1.** *Under the assumptions of Theorem 2, if all directed paths from any ancestor of $P$ to $X$ in the graph $\mathcal{G}$ are blocked by $P$, then any predictor based on the* adjusted features $\tilde{X} := X - \mathbb{E}[X \mid P]$ *exhibits no proxy discrimination and can be learned from the observational distribution $\mathbb{P}(P, X, Y)$ when target labels $Y$ are available.*

Our definition of proxy discrimination in expectation (4) is motivated by a weaker notion proposed in [24]. It asks for the expected outcome to be the same across the different populations $\mathbb{E}[R \mid P = p] = \mathbb{E}[R \mid P = p']$. Again, when talking about proxies, we must be careful to distinguish conditional and interventional expectations, which is captured by the following proposition and its corollary.

**Proposition 3.** *Any predictor of the form $R = \lambda(X - \mathbb{E}[X \mid do(P)]) + c$ for $\lambda, c \in \mathbb{R}$ exhibits no proxy discrimination in expectation.*

From this and the proof of Corollary 1 we conclude the following Corollary.

**Corollary 2.** *If all directed paths from any ancestor of $P$ to $X$ are blocked by $P$, any predictor of the form $R = r(X - \mathbb{E}[X \mid P])$ for linear $r$ exhibits no proxy discrimination in expectation and can be learned from the observational distribution $\mathbb{P}(P, X, Y)$ when target labels $Y$ are available.*

# 5 Conclusion

The goal of our work is to assay fairness in machine learning within the context of causal reasoning. This perspective naturally addresses shortcomings of earlier statistical approaches. Causal fairness criteria are suitable whenever we are willing to make assumptions about the (causal) generating

process governing the data. Whilst not always feasible, the causal approach naturally creates an incentive to scrutinize the data more closely and work out plausible assumptions to be discussed alongside any conclusions regarding fairness.

Key concepts of our conceptual framework are *resolving variables* and *proxy variables* that play a dual role in defining causal discrimination criteria. We develop a practical procedure to remove proxy discrimination given the structural equation model and analyze a similar approach for unresolved discrimination. In the case of proxy discrimination for linear structural equations, the procedure has an intuitive form that is similar to heuristics already used in the regression literature. Our framework is limited by the assumption that we can construct a valid causal graph. The removal of proxy discrimination moreover depends on the functional form of the causal dependencies. We have focused on the conceptual and theoretical analysis, and experimental validations are beyond the scope of the present work.

The causal perspective suggests a number of interesting new directions at the technical, empirical, and conceptual level. We hope that the framework and language put forward in our work will be a stepping stone for future investigations.

## Footnotes

[1]As it is not needed in our work, we do not discuss the graph-theoretic notion of d-separation.

[2]If we do assume the Markov condition and faithfulness, then conditional independences determine the graph up to its so called *Markov equivalence class*.

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
