[Supplementary Material]

# Avoiding Discrimination through Causal Reasoning

**Niki Kilbertus**[†‡]
nkilbertus@tue.mpg.de

**Mateo Rojas-Carulla**[†‡]
mrojas@tue.mpg.de

**Giambattista Parascandolo**[†§]
gparascandolo@tue.mpg.de

**Moritz Hardt**[*]
hardt@berkeley.edu

**Dominik Janzing**[†]
janzing@tue.mpg.de

**Bernhard Schölkopf**[†]
bs@tue.mpg.de

[†]Max Planck Institute for Intelligent Systems
[‡]University of Cambridge
[§]Max Planck ETH Center for Learning Systems
[*]University of California, Berkeley

## Supplementary material

### Proof of Theorem 1

**Theorem.** *Given a joint distribution over the protected attribute $A$, the true label $Y$, and some features $X_1, \ldots, X_n$, in which we have already specified the resolving variables, no observational criterion can generally determine whether the Bayes optimal unconstrained predictor or the Bayes optimal equal odds predictor exhibit unresolved discrimination.*

*Proof.* Let us consider the two graphs in Figure 2. First, we show that these graphs can generate the same joint distribution $\mathbb{P}(A, Y, X_1, X_2, R^*)$ for the Bayes optimal unconstrained predictor $R^*$.

We choose the following structural equations for the graph on the left[1]

- $A = Ber(1/2)$
- $X_1$ is a mixture of Gaussians $\mathcal{N}(A + 1, 1)$ with weight $\sigma(2A)$ and $\mathcal{N}(A - 1, 1)$ with weight $\sigma(-2A)$
- $Y = Ber(\sigma(2X_1))$
- $X_2 = X_1 - A$
- $R^* = X_1$
- $(\tilde{R} = X_2)$

where the Bernoulli distribution $Ber(p)$ without a superscript has support $\{-1, 1\}$.

For the graph on the right, we define the structural equations

- $A = Ber(1/2)$
- $Y = Ber(\sigma(2A))$
- $X_2 = \mathcal{N}(Y, 1)$

- $X_1 = A + X_2$
- $R^* = X_1$
- $(\tilde{R} = X_2)$

First we show that in both scenarios $R^*$ is actually an optimal score. In the first scenario $Y \perp\!\!\!\perp A \mid X_1$ and $Y \perp\!\!\!\perp X_2 \mid X_1$ thus the optimal predictor is only based on $X_1$. We find

$$\Pr(Y = y \mid X_1 = x_1) = \sigma(2x_1 y) , \tag{1}$$

which is monotonic in $x_1$. Hence optimal classification is obtained by thresholding a score based only on $R^* = X_1$.

In the second scenario, because $Y \perp\!\!\!\perp X_1 \mid \{A, X2\}$ the optimal predictor only depends on $A, X_2$. We compute for the densities

$$\mathbb{P}(Y \mid X_2, A) = \frac{\mathbb{P}(Y, X_2, A)}{\mathbb{P}(X_2, A)} \tag{2a}$$

$$= \frac{\mathbb{P}(X_2, A \mid Y)\mathbb{P}(Y)}{\mathbb{P}(X_2, A)} \tag{2b}$$

$$= \frac{\mathbb{P}(X_2 \mid Y)\mathbb{P}(A \mid Y)\mathbb{P}(Y)}{\mathbb{P}(X_2, A)} \tag{2c}$$

$$= \frac{\mathbb{P}(X_2 \mid Y)\frac{\mathbb{P}(Y \mid A)\mathbb{P}(A)}{\mathbb{P}(Y)}\mathbb{P}(Y)}{\mathbb{P}(X_2, A)} \tag{2d}$$

$$= \frac{\mathbb{P}(X_2 \mid Y)\mathbb{P}(Y \mid A)\mathbb{P}(A)}{\mathbb{P}(X_2, A)} , \tag{2e}$$

where for the third equal sign we use $A \perp\!\!\!\perp X_2 \mid Y$. In the numerator we have

$$\mathbb{P}(X_2 \mid Y = y)(x_2)\mathbb{P}(Y \mid A = a)(y)\mathbb{P}(A)(a) = f_{\mathcal{N}(y,1)}(x_2) f_{Ber(\sigma(2a))}(y) f_{Ber(1/2)}(a) , \tag{3}$$

where $f_D$ is the probability density function of the distribution $D$. The denominator can be computed by summing up (15) for $y \in \{-1, 1\}$. Overall this results in

$$\Pr(Y = y \mid X_2 = x_2, A = a) = \sigma(2y(a + x_2)) .$$

Since by construction $X_1 = A + X_2$, the optimal predictor is again $R^* = X_1$. If the joint distribution $\mathbb{P}(A, Y, R^*)$ is identical in the two scenarios, so are the joint distributions $\mathbb{P}(A, Y, X_1, X_2, R^*)$, because of $X_1 = R^*$ and $X_2 = X_1 - A$.

To show that the joint distributions $\mathbb{P}(A, Y, R^*) = \mathbb{P}(Y \mid A, R^*)\mathbb{P}(R^* \mid A)\mathbb{P}(A)$ are the same, we compare the conditional distributions in the factorization.

Let us start with $\mathbb{P}(Y \mid A, R^*)$. Since $R^* = X_1$ and in the first graph $Y \perp\!\!\!\perp A \mid X_1$, we already found the distribution in (13). In the right graph, $\mathbb{P}(Y \mid R^*, A) = \mathbb{P}(Y \mid X_2 + A, A) = \mathbb{P}(Y \mid X_2, A)$ which we have found in (14) and coincides with the conditional in the left graph because of $X_1 = A + X_2$.

Now consider $R^* \mid A$. In the left graph we have $\mathbb{P}(R^* \mid A) = \mathbb{P}(X_1 \mid A)$ and the distribution $\mathbb{P}(X_1 \mid A)$ is just the mixture of Gaussians defined in the structural equation model. In the right graph $R^* = A + X_2 = Y + \mathcal{N}(A, 1)$ and thus $\mathbb{P}(R^* \mid A) = \mathcal{N}(A \pm 1)$ for $Y = \pm 1$. Because of the definition of $Y$ in the structural equations of the right graph, following a Bernoulli distribution with probability $\sigma(2A)$, this is the same mixture of Gaussians as the one we found for the left graph.

Clearly the distribution of $A$ is identical in both cases.

Consequently the joint distributions agree.

When $X_1$ is an resolving variable, the optimal predictor in the left graph does not exhibit unresolved discrimination, whereas the graph on the right does.

The proof for the equal odds predictor $\tilde{R}$ is immediate once we show $\tilde{R} = X_2$. This can be seen from the graph on the right, because here $X_2 \perp\!\!\!\perp A \mid Y$ and both using $A$ or $X_1$ would violate the equal odds condition. Because the joint distribution in the left graph is the same, $\tilde{R} = X_2$ is also the optimal equal odds score. $\qquad\square$

**Proof of Proposition 1**

**Proposition.** *If there is no directed path from a proxy to a feature, unawareness avoids proxy discrimination.*

*Proof.* An unaware predictor $R$ is given by $R = r(X)$ for some function $r$ and features $X$. If there is no directed path from proxies $P$ to $X$, i.e. $P \notin ta^{\mathcal{G}}(X)$, then $R = r(X) = r(ta^{\mathcal{G}}(X)) = r(ta_P^{\mathcal{G}}(X))$. Thus $\mathbb{P}(R \mid do(P = p)) = \mathbb{P}(R)$ for all $p$, which avoids proxy discrimination. $\square$

**Proof of Theorem 2**

**Theorem.** *Let the influence of $P$ on $X$ be additive and linear, i.e.*
$$X = f_X(P, ta_P^{\mathcal{G}}(X)) = g_X(ta_P^{\mathcal{G}}(X)) + \mu_X P$$
*for some function $g_X$ and $\mu_X \in \mathbb{R}$. Then any predictor of the form*
$$R = r(X - \mathbb{E}[X \mid do(P)])$$
*for some function $r$ exhibits no proxy discrimination.*

*Proof.* It suffices to show that the argument of $r$ is constant w.r.t. to $P$, because then $R$ and thus $\mathbb{P}(R)$ are invariant under changes of $P$. We compute
$$\begin{aligned}
\mathbb{E}[X \mid do(P)] &= \mathbb{E}[g_X(ta_P^{\mathcal{G}}(X)) + \mu_X P \mid do(P)] \\
&= \underbrace{\mathbb{E}[g_X(ta_P^{\mathcal{G}}(X)) \mid do(P)]}_{=0} + \mathbb{E}[\mu_X P \mid do(P)] \\
&= \mu_X P \, .
\end{aligned}$$
Hence,
$$X - \mathbb{E}[X \mid do(P)] = g_X(ta_P^{\mathcal{G}}(X))$$
is clearly constant w.r.t. to $P$. $\square$

**Proof of Corollary 1**

**Corollary.** *Under the assumptions of Theorem 2, if all directed paths from any ancestor of $P$ to $X$ in the graph $\mathcal{G}$ are blocked by $P$, then any predictor based on the* adjusted features $\tilde{X} := X - \mathbb{E}[X \mid P]$ *exhibits no proxy discrimination and can be learned from the observational distribution $\mathbb{P}(P, X, Y)$ when target labels $Y$ are available.*

*Proof.* Let $Z$ denote the set of ancestors of $P$. Under the given assumptions $Z \cap ta^{\mathcal{G}}(X) = \emptyset$, because in $\mathcal{G}$ all arrows into $P$ are removed, which breaks all directed paths from any variable in $Z$ to $X$ by assumption. Hence the distribution of $X$ under an intervention on $P$ in $\tilde{\mathcal{G}}$, where the influence of potential ancestors of $P$ on $X$ that does not go through $P$ would not be affected, is the same as simply conditioning on $P$. Therefore $\mathbb{E}[X \mid do(P)] = \mathbb{E}[X \mid P]$, which can be computed from the joint observational distribution, since we observe $X$ and $P$ as generated by $\tilde{\mathcal{G}}$. $\square$

**Proof of Proposition 3**

**Proposition.** *Any predictor of the form $R = \lambda(X - \mathbb{E}[X \mid do(P)]) + c$ for linear $\lambda, c \in \mathbb{R}$ exhibits no proxy discrimination in expectation.*

*Proof.* We directly test the definition of proxy discrimination in expectation using the linearity of the expectation
$$\begin{aligned}
\mathbb{E}[R \mid do(P = p)] &= \mathbb{E}[\lambda(X - \mathbb{E}[X \mid do(P)]) + c \mid do(P = p)] \\
&= \lambda(\mathbb{E}[X \mid do(P = p)] - \mathbb{E}[X \mid do(P = p)]) + c \\
&= c \, .
\end{aligned}$$
This holds for any $p$, hence proxy discrimination in expectation is achieved. $\square$

**Additional statements**

Here we provide an additional statement that is a first step towards the "opposite direction" of Theorem 2, i.e. whether we can infer information about the structural equations, when we are given a predictor of a special form that does not exhibit proxy discrimination.

**Theorem.** *Let the influence of $P$ on $X$ be additive and linear and let the influence of $P$ on the argument of $R$ be additive linear, i.e.*

$$f_X(ta^{\mathcal{G}}(X)) = g_X(ta_P^{\mathcal{G}}(X)) + \mu_X P$$
$$f_R(P, ta_P^{\mathcal{G}}(X)) = h(g_R(ta_P^{\mathcal{G}}(X)) + \mu_R P)$$

*for some functions $g_X, g_R$, real numbers $\mu_X, \mu_R$ and a smooth, strictly monotonic function $h$. Then any predictor that avoids proxy discrimination is of the form*

$$R = r(X - \mathbb{E}[X \,|\, do(P)])$$

*for some function $r$.*

*Proof.* From the linearity assumptions we conclude that

$$\hat{f}_R(P, X) = h(g_X(ta_P^{\mathcal{G}}(X)) + \mu_X P + \hat{\mu}_R P) \,,$$

with $\hat{\mu}_R = \mu_R - \mu_P$ and thus $g_X = g_R$. That means that both the dependence of $X$ on $P$ along the path $P \to \cdots \to X$ as well as the direct dependence of $R$ on $P$ along $P \to R$ are additive and linear.

To avoid proxy discrimination, we need

$$\mathbb{P}(R \,|\, do(P = p)) = \mathbb{P}(h(g_R(ta_P^{\mathcal{G}}(X)) + \mu_R p)) \tag{4a}$$

$$\overset{!}{=} \mathbb{P}(h(g_R(ta_P^{\mathcal{G}}(X)) + \mu_R p')) = \mathbb{P}(R \,|\, do(P = p')) \,. \tag{4b}$$

Because $h$ is smooth an strictly monotonic, we can conclude that already the distributions of the argument of $h$ must be equal, otherwise the transformation of random variables could not result in equal distributions, i.e.

$$\mathbb{P}(g_R(ta_P^{\mathcal{G}}(X)) + \mu_R p) \overset{!}{=} \mathbb{P}(g_R(ta_P^{\mathcal{G}}(X)) + \mu_R p') \,.$$

Since, up to an additive constant, we are comparing the distributions of the *same* random variable $g_R(ta_P^{\mathcal{G}}(X))$ and not merely identically distributed ones, the following condition is not only sufficient, but also necessary for (16)

$$g_R(ta_P^{\mathcal{G}}(X)) + \mu_R p \overset{!}{=} g_R(ta_P^{\mathcal{G}}(X)) + \mu_R p' \,.$$

This holds true for all $p, p'$ only if $\mu_R = 0$, which is equivalent to $\hat{\mu}_R = -\mu_P$.

Because as in the proof of 2

$$\mathbb{E}[X \,|\, do(P)] = \mu_X P,$$

under the given assumptions any predictor that avoids proxy discrimination is simply

$$R = X + \mu_R P = X - \mathbb{E}[X \,|\, do(P)] \,.$$

$\square$

## Footnotes

[1]$\sigma(x) = 1/(1 + e^{-x})$