[Reviews · NeurIPS 2017]

Reviewer 1



This paper formulates fairness for protected attributes as a causal inference problem. It is a well-written paper that makes an important connection that discrimination, in itself, is a causal concept that can benefit from using causal inference explicitly. The major technical contribution of the paper is to show a method using causal graphical models for ensuring fairness in a predictive model. In line with social science research on discrimination, they steer away from thinking about a gender or race counterfactual which can be an unwieldy or even absurd counterfactual to construct---what would happen if someone's race is changed---and instead focus on using proxy variables and controlling causal pathways from proxy variables to an outcome. This is an important distinction, and allows us to declare proxy variables such as home location or name and use them for minimizing discrimination. To avoid using this proxy variable for prediction, the proposed method first assumes a likely graphical model for all relevant variables and then uses parametric assumptions to cancel out the effect of proxies. While the authors make unique theoretical contributions in connecting do-calculus interventions to fairness, in practice, this reveals limitations in using their method: first, we need to be able to construct a valid causal model. This can be hard in practical scenarios of race or gender discrimination when there can be many variables that affect an individual's outcome, and almost all of them may be connected to each other and to the outcome. Second, the statistical correction for proxies depends critically on the assumed functional form of causal relationships, which again can be arbitrary. I would encourage the authors to discuss these limitations briefly, if possible. That said, every other method (even non-causal) suffers from at least one of these limitations implicitly, so in comparison this can be a sound strategy to employ. Further, laying down the causal model enables the research community and policy makers to be explicit about their assumptions, which can enhance clarity and limitations of any predictive method. In addition, the authors give a clear (though artificial) example of how causal assumptions are necessary for detecting discrimination: they show that two different causal models can lead to the same observed data distribution, even though only one of them is a discriminatory causal model. I found this quite insightful.

Reviewer 2



The authors point out that a causal inference viewpoint is a good way to view algorithmic discrimination and fairness, and then carry through this viewpoint to note its various insights that are unavailable via observational data analysis through standard machine learning and data mining. This is a good piece of work, with good writing, correct math, and important start to an area of study that clearly needs to be studied. There is a bit too much material of a tutorial nature. The authors have not cited https://arxiv.org/abs/1608.03735.

Reviewer 3



The paper proposes a causal view on the question of fair machine learning. Its main contribution is the introduction of causal language to the problem, and specifically the notion of "resolving variable": a variable that mediates the causal effect of a protected attribute in a way the user deems "fair". An example would be the variable "choice of departments" in the famous Simpson's paradox college-admission scenario. Overall I think this paper has a lot of potential, but still needs some work on clarifying many of the concepts introduced. Fairness is a difficult subject as it clearly goes beyond the math and into societal and normative discussions. As such, I am looking forward to the authors replies to my comments and questions below, hoping to have a fruitful discussion. Pros: 1. This paper present an important contribution to the discussion on fairness, by introducing the notion of resolving variable. I have already found this term useful in discussions about the subject. 2. The paper shows how in relatively simple scenarios the proposed definitions and methods reduce to very reasonable solutions, for example the result in Theorem 2 and its corollaries. Cons: 1. No experimental results, not even synthetic. Other papers in the field have experiments, e.g. Hardt Price & Srebro (2016) and Chulechova (2016). 2. Lack of clarity in distinction between proxies and non-resolving variables. 3. I am not convinced of the authors view on the role of proxy variables. 4. I am not convinced that some of the proposed definitions capture the notion of fairness the authors claim to capture. Specific comments: 1. What exactly is the difference between a proxy variable and a non-resolving variable? Does the difference lie only in the way they are used? 2. Why is the definition of proxy discrimination (def. 3) the one we want? Wouldn't the individual proxy discrimination be the one which more closely correlates with our ideas of fairness? E.g., say name (variable P) affects past employment (variable X), which in turn affects the predictor. Couldn't averaging this effect over X's (even considering the do-operator version) lead to a predictor which is discriminating with respect to a specific X? 3. Why is the notion of proxy variables the correct one to control for? I think on the contrary, in many cases we *do* have access to the protected attribute (self-declared race, gender, religion etc.). It's true that conceiving intervention on these variables is difficult, but conceiving the paths through which these variables affect outcomes is more plausible. The reason gender affects employment goes beyond names: it relates to opportunities presented, different societal attitudes, and so far. The name is but one path (an unresolved path) through which gender can affect employment. 4. The wording in the abstract: "what do we want to assume about the causal DGP", seems odd. The causal DGP exists in the world, and we cannot in general change it. Do the authors mean the causal DGP *that gives rise to a predictor R* ? 5. Why have the mentioned approaches to individual fairness not gained traction? 6. The paper would be better if it mentioned similar work which recently appeared on arxiv by Nabi & Shpitser, "Fair Inference on Outcomes". In general I missed a discussion of mediation, which is a well known term in causal inference and seems to be what this paper is aiming at understanding. 7. I am confused by the relation between the predictions R and the true outcomes Y. Shouldn't the parameters leading to R be learned from data which includes Y ? 8. In figure 4, why does the caption say that "we generically do not want A to influence R"? Isn't the entire discussion premised on the notion that sometimes it's ok if A influences R, through a resolving variable? Minor comments: 1. line 215 I think there's a typo, N_A should be N_P?